# Haptic Glove and Platform with Gestural Control For Neuromorphic Tactile Sensory Feedback In Medical Telepresence [note 1]

**DOI:** 10.3390/s19030641

**Published:** 2019-02-03

**Authors:** Jessica D’Abbraccio, Luca Massari, Sahana Prasanna, Laura Baldini, Francesca Sorgini, Giuseppe Airò Farulla, Andrea Bulletti, Marina Mazzoni, Lorenzo Capineri, Arianna Menciassi, Petar Petrovic, Eduardo Palermo, Calogero Maria Oddo

**Affiliations:** 1The BioRobotics Institute, Sant’Anna School of Advanced Studies, 56025 Pisa, Italy; jessica.dabbraccio@santannapisa.it (J.D.); luca.massari@santannapisa.it (L.M.); sahana.prasanna@santannapisa.it (S.P.); frasorg@gmail.com (F.S.); arianna.menciassi@santannapisa.it (A.M.); 2Department of Linguistics and Comparative Cultural Studies, Ca’ Foscari University of Venice, 30123 Venice, Italy; 3Department of Mechanical and Aerospace Engineering, “Sapienza” University of Rome, 00185 Rome, Italy; laurabaldini93@gmail.com (L.B.); eduardo.palermo@uniroma1.it (E.P.); 4Department of Control and Computer Engineering, Politecnico di Torino, 10129 Turin, Italy; giuseppe.airof@gmail.com; 5Department of Information Engineering, Università degli studi di Firenze, 50121 Florence, Italy; andrea.bulletti@unifi.it (A.B.); m.mazzoni@ifac.cnr.it (M.M.); lorenzo.capineri@unifi.it (L.C.); 6Istituto di Fisica Applicata “Nello Carrara”, Consiglio Nazionale delle Ricerche of Italy, 50019 Sesto Fiorentino, Italy; 7Production Engineering Department, Faculty of Mechanical Engineering, University of Belgrade, 11120 Belgrade, Serbia; pbpetrovic@mas.bg.ac.rs; 8Academy of Engineering Sciences of Serbia (AISS), 11120 Belgrade, Serbia

**Keywords:** nodules detection, neuromorphic touch, polymeric phantom, sensory augmentation, tactile telepresence, teleoperation, tele-palpation, vibro-tactile stimulation

## Abstract

Advancements in the study of the human sense of touch are fueling the field of haptics. This is paving the way for augmenting sensory perception during object palpation in tele-surgery and reproducing the sensed information through tactile feedback. Here, we present a novel tele-palpation apparatus that enables the user to detect nodules with various distinct stiffness buried in an ad-hoc polymeric phantom. The contact force measured by the platform was encoded using a neuromorphic model and reproduced on the index fingertip of a remote user through a haptic glove embedding a piezoelectric disk. We assessed the effectiveness of this feedback in allowing nodule identification under two experimental conditions of real-time telepresence: In Line of Sight (ILS), where the platform was placed in the visible range of a user; and the more demanding Not In Line of Sight (NILS), with the platform and the user being 50 km apart. We found that the entailed percentage of identification was higher for stiffer inclusions with respect to the softer ones (average of 74% within the duration of the task), in both telepresence conditions evaluated. These promising results call for further exploration of tactile augmentation technology for telepresence in medical interventions.

## 1. Introduction

### 1.1. Motivation and Challenge Definition

The impact of haptic devices has grown tremendously over the last decade [1,2,3]. These devices connect the user to a virtual object, providing tactile feedback directly to the human skin. Through the sense of touch humans are able to gather a wide range of information about the surrounding environment. Thus, the delivery of tactile information is crucial for sampling objects, in particular when they are occluded from sight. Haptic devices pave the way for most of the telepresence applications based on the sense of touch as a communication channel. However, if compared to the advanced technologies that are available commercially to convey vision and auditory information with a very high level of fidelity, tactile telepresence technologies are still in a pioneering phase. The present study addresses this challenge with specific reference to potential applications in medical robotics, by proposing a markerless gesture-based controller of a mechatronic platform with tactile sensory feedback delivered to a user seeking remote buried nodules. This task is selected as a benchmarking reference of the developed system because palpation is a very important practice in medical diagnosis and surgical intervention.

### 1.2. Related Work and State of the Art

The development of technologies inspired by the study of human sense of touch is being contributed to by the integration of several research fields, such as biomedical engineering, robotics and biorobotics, measurement and instrumentation, computer graphics, cognitive science, neuroscience, and psychophysics. Research on human tactile sensing has characterized the role of functionally diverse skin receptors [4,5,6,7] densely populating the fingertips (up to hundreds of units per square centimeter). Biological mechanoreceptors allow for a high resolution of sensed information, and encode a wide range of temporal and spatial stimulations [8,9,10,11].

Artificial tactile sensors and haptic interfaces are the two parts (remote and close to the operator, respectively) of a tactile telepresence application. Considering the development of tactile sensors and artificial skins [12,13], applications of tactile sensing include tactile-based material recognition [14,15], tactile based object manipulation, and slip detection [16,17,18].

Haptic devices could contribute to the performing of different tasks in various scenarios: in rehabilitation procedures, to develop tactile aids for visual and auditory sensory disabled persons [19,20,21,22,23]; and in telepresence operations, to improve user perception capabilities through vibro-tactile feedback [24,25,26,27,28]. Beyond these applications, tactile technologies employed in telepresence and teleoperation scenarios have been widely used in the field of minimally invasive surgery (MIS) and robotic minimally invasive surgery (RMIS), to overcome the absence of both tactile and force feedback conveyed to the physician. Feedback is crucial for helping the surgeon to preserve healthy tissues, as well as for detecting differences in stiffness throughout the palpated sites [29,30,31,32,33,34,35]. Despite the growth in recent interest and research, the degree of maturity of touch sensing is still lagging behind that of other perceptive technologies, such as audio or computer vision [36,37,38,39,40,41]. A plausible explanation is in the biological complexity of the sense of touch, where the distributed sensitized region covers the entire body [42], while other human senses have localized sensitive areas.

In a previous work [43], we developed a mechatronic platform interfaced with a vibro-tactile glove for tactile augmentation in telepresence under passive exploration of remote stimuli. The results demonstrated the efficiency of the system in presenting mechanical information about test objects. The stiffness of different materials, converted into spikes with proper frequency through a neural model, was used to drive piezoelectric disks embedded in the index and the thumb of a vibro-tactile glove. The designed psychophysical protocol for discriminating stiffness, tested with 2-Alternative Forced Choice passive touch, unveiled specific perceptual thresholds, derived using the Ulrich–Miller Cumulative Distribution Function. Although passive touch conveys information about the miscellaneous properties of the explored object, an active exploration permits enrichment of the tactile signals, thanks to integration with the proprioceptive inputs [44,45,46,47]. Remote tele-palpation under active touch with gesture-based control and spike-based feedback is the objective of the present study.

### 1.3. Contribution of the Present Study

Building on the findings of the previous study, here we propose a telepresence system with active exploration of a silicon phantom that embeds elements with different stiffness within. In this condition, the user can directly control and move the indenting platform, with an additional optical sensor for tracking the hand movements. We investigated two experimental conditions of telepresence: (i) ILS, where the platform was placed In Line of Sight of the user [48]; (ii) NILS, where the platform was Not In Line of Sight, in a different location with respect to the user, for a more challenging task. Hence, the objective of the present study is to investigate the mechanisms of tactile perception under active gestural control and the effectiveness of the proposed spike-based feedback strategy. In particular, the delivered vibro-tactile feedback mimics the language of tactile receptors, generated by a neuromorphic model [49,50,51].

The present work is organized as follows: Section 2 is partitioned into (i) Experimental Setup; (ii) Platform and Inclusions Characterization Protocols; and (iii) Psychophysical Experiments sub-sections. In these sections, the telepresence system with the related two sub-systems and the used phantom are described and details about the performed characterizations and the performed protocols are given. The results are reported in Section 3, while Section 4 deals with the discussion and conclusions of the present study.

## 2. Materials and Methods

### 2.1. Experimental Setup

The experimental apparatus was composed of two essential sub-setups, positioned at a proper distance for achieving telepresence conditions (Figure 1A).

The first sub-setup consisted of a piezoelectric disk (7BB-12-9, MuRata, Kyoto Prefecture, Japan) encapsulated in silicone rubber, with a customized process, which was placed on the index fingertip of a textile glove to deliver the feedback [52]. An optical sensor (Leap Motion, CA, USA) tracked the user’s hand gesture. We defined as haptic sub-system this first sub-setup located in a laboratory of The BioRobotics Institute of Sant’Anna School of Advanced Studies, Pontedera (Pisa, Italy). A graphical user interface (GUI) was developed using LabVIEW (National Instruments Corp., USA) for acquiring speed and position of the hand’s center of mass, handling communication with the other remote sub-setup, and recording hand position data.

We defined as tactile sub-system, the second remote sub-setup used for the exploration of the phantom. This platform included: a cartesian manipulator (X–Y and Z, 8MTF-102LS05 and 8MVT120-25- 4247, STANDA, Vilnius, Lithuania); a 6-axis load cell (Nano 43, ATI Industrial Automation, Apex, USA) to measure contact forces between a customized indenter with a spherical tip of 3 mm radius, mounted on the load cell; and the phantom during the active sliding. A second GUI was designed for real-time control of the motorized stages and data communication.

In the ILS session, the tactile sub-system was placed near the user to perform experiments in streamlined telepresence. In the NILS session, instead, it was located in a remote laboratory in Florence, Italy, which was about 50 km apart from the haptic sub-system, thus increasing the challenge of the proposed task. Data communication between the two sub-setups was provided through the User Datagram Protocol (UDP) that ensured a maximum latency of 15 ms. The adopted communication protocol allowed bidirectional streaming of data: hand gestures from the haptic sub-system to the mechatronic platform, and normal force from the tactile sub-system to the glove to be encoded and then delivered (Figure 2).

During experiments, the user actively explored and searched for stiffer areas in the polymeric phantom. Four different rubber materials were used to cast 12 hemispherical inclusions (3 replicas of each material), of 5 mm radius, randomly inserted across the X–Y plane in a silicon cuboidal block, 100 × 100 × 15 mm^3^ in size. The chosen polymers for the inclusions were: Sorta Clear 40 (Smooth-on, PA, USA) as the stiffest, polydimethylsiloxane (PDMS) Sylgard 184 (Dow Corning, USA), Dragon Skin 30, and Dragon Skin 20, while Dragon Skin 10, the softest, was used for the cuboidal block. 

The platform control during the experimental session was based on the user’s hand movements, tracked by the 3-D optical sensor. The infrared light-based gesture sensor used had a field of view of about 150 degrees and a range between 25 and 600 mm above the device, and it showed proper performance when it had a high-contrast view of the hand to be detected and tracked. Therefore, the device was placed just below and close to the rest position of the user’s hand, and the workspace was set according to the specifications of the gesture sensor. Details about the reference coordinates of the gesture sensor, highlighting the rest position and volume, are shown in Figure 1B. As the user’s hand moved out of this volume, the motorized stages followed along the same direction at a speed proportional to the displacement of the user’s hand. The assigned speed was calculated using the difference ρ − ρ_0_, where ρ was the distance between the hand center of mass and the center of the sensor, and ρ_0_ was the radius of the neutral spherical region, set to 50 mm. The user was provided with visual feedback about the position of the indenter on the phantom in the remote environment, without any information on the location of the inclusions.

The contact between the indenter tip and the polymeric test object generated a force. To avoid mechanical damage to the phantom and load cell due to the user’s upward movement, a force threshold (0.5 N) was introduced. The measured force was sent to the haptic sub-system to be encoded into spike patterns, which triggered the piezoelectric actuator. We implemented a neuromorphic feedback strategy based on a regular Izhikevich spiking model discretized using Euler’s method at 5 kHz [53]. The chosen model efficiently encodes the temporal dynamics of a mechanoreceptor including the biological plausibility of the computationally intense Hodgkin–Huxley model [54] and the computational efficiency of the integrate-and-fire model [55]. 

The neuromorphic activation of the transducer was achieved by setting the input to the neuron proportional to the magnitude of the normal force measured by the remote subsystem. Further details about the neural model and a definition of its parameters can be found in our previous works [43,48]. Initial calibrations were also performed to counterbalance the effect of saturation of the neural model [48]. 

The spikes generated by the Izhikevich model were sent to the vibro-tactile glove by means of a piezoelectric driver (DRV2667, Texas Instruments). This driver facilitated the setting of the actuation parameters in analog mode to have a gain of 40.7 dB, 200 V peak-to-peak voltage amplitude, and 105 V offset voltage.

### 2.2. Platform and Inclusions Characterization Protocols

A characterization protocol was performed aiming at assessing the uncertainty of the apparatus in tracking and reproducing the user’s hand pose. Five subjects (4 men and 1 woman between 28 and 30 years of age), enrolled among the staff of The BioRobotics Institute of Sant’Anna School of Advanced Studies, took part in the experiment. They were asked to drive the stages of the platform in order to perform a set of target trajectories within the X–Y cartesian plane. The target trajectory and the subject’s relative position on this plane were part of the GUI providing visual feedback in the haptic sub-system. The cluster consisted of 2 square-shaped (s_1_= 60 mm and s_2_ = 30 mm, in length) and 2 circular (r_1_ = 30 mm and r_2_ = 15 mm, in radius) trajectories, which had to be followed 3 times by each subject, starting from their centers. Deviations in the trajectories, used to evaluate the system in terms of performance, were calculated using the area comprised between the perimeter of the target and executed trajectories, by means of the boundary function (MATLAB, The MathWorks, Inc., Natick, Massachusetts, USA. The error rates between the tracked area and target were calculated as their ratio and are reported in Section 3.

Moreover, before involving human subjects, the phantom was mechanically characterized to assess the vertical stiffness (ΔF_z_/Δz) of the nodules and the surrounding soft material by means of the proposed platform equipped with a flat indenter. The adopted experimental protocol consisted of 5 trials in which the entire set of inclusions experienced an indentation at a fixed force threshold (F_z_ = 0.5 N) and speed (v = 0.125 mms^−1^). To estimate the stiffness of the investigated polymers, the vertical component of the force collected during each compression was processed through scripts in MATLAB and results of such a characterization are reported in Section 3.

### 2.3. Inclusions Identification Experimental Methods

Psychophysical experiments were performed to validate the proposed system for delivering stiffness information about the palpated nodules in both ILS and NILS tactile telepresence conditions. The experiments involved 10 participants (7 men and 3 women between 24 and 33 years of age) in ILS, and 15 (9 men and 6 women between 25 and 37 years of age) in NILS, enrolled among the university students or staff of The BioRobotics Institute of Sant’Anna School of Advanced Studies, Pisa, Italy. The participants took a comfortable posture at the control workstation in the laboratory, where the haptic sub-system was located. They wore the vibro-tactile glove on their dominant hand and a headset to eliminate environmental noise. To familiarize participants with the driving of the tactile platform, each participant took part in a fifteen minute training session. Moreover, this preliminary task got the users accustomed to the vibro-tactile signal exerted by the piezoelectric actuator of the glove. Answers provided during the training sessions were not included in the analyzed results. Both ILS and NILS psychophysical experiments consisted of a tactile identification task: within six minute time-period of the protocol, the participants unreservedly explored the silicon block and pressed a button on a keyboard on any occasion of perceived frequency variation in the vibro-tactile feedback.

The performances of the participants were evaluated in MATLAB in terms of rate of correct identification of the inclusions, using parameters calculated for both ILS and NILS populations and for each participant. Specifically, we evaluated: (a) the number of true positives (TP), (b) the number of false positives (FP), and (c) accuracy (TP/P, with P = collected responses—FP). These parameters were computed as a function of the center-to-center distance between the position of the perceived inclusions and the nearest actual inclusion. If the perceived position was felt to be within a radius of 10 mm from the center of the nearest inclusion (i.e., the distance was equal to the diameter of the inclusion), the perceived inclusion was classified as TP; otherwise it was classified as FP. The classification of collected responses was also evaluated for a lower tolerance, 5 mm, and two greater ones, 15 mm and 20 mm. A logistic fitting, using a Cumulative Distribution Function (CDF) [56] and the *nlinfit* MATLAB function, was performed for each material to carry out the rate of correct perception and the perceptual thresholds for both the investigated experimental conditions.

## 3. Results

### 3.1. Platform and Inclusions Characterization Results

In the platform characterization, the trajectories followed by each user for individual tests are shown in Figure 3. The target areas, the tracked areas, their difference, and the error rates for all the presented trajectories are reported in Table 1 in terms of mean and standard deviation amongst subjects. Based on the user’s gestures, the platform was able to easily follow square geometries (error rate lower than 5%) as well as circular trajectories (but with higher average errors, comprised between 5% and 13%, being the task more challenging than tracking square geometries). The error rate increased with the area being explored, presumably due to non-constant spatial resolution of the gesture-sensor.

In the phantom characterization, the stiffness of each material was measured by assuming the polymers had a linear response in the range of applied forces. According to the operated characterization, the stiffer materials were Sorta Clear (3.69 N·mm^−1^) and PDMS (3.68 N·mm^−1^), while Dragon Skin 30 (2.88 N·mm^−1^), Dragon Skin 20 (2.74 N·mm^−1^), and Dragon Skin 10 (2.14 N·mm^−1^) showed lower values (Figure 4).

### 3.2. Inclusions Identification Experimental Results

In the psychophysical experiments with tactile feedback in ILS and NILS telepresence conditions, we successfully delivered a neuromorphic stimulation encoding the stiffness of the telepalpated phantom. The inclusions recognized by the user were recorded with a key press. An example of the path followed and the responses are shown in Figure 5.

Results show that an average of 63% and 60% of the inclusions were correctly perceived and declared within a tolerance of 10 mm to the nearest inclusion, for ILS and NILS experiments, respectively. In both cases, with a tolerance higher than the inclusion diameter (i.e., >10 mm), the rate of identified inclusions was slightly higher (Figure 6).

The identification rate for all the participants, across populations, was evaluated for each material and is presented in Figure 7 with increasing stiffness along with the interquartile range (IQR) for the accuracy of declared inclusions. Performance, in terms of correct identification of inclusions, was 65% and 70% for stiffer stimuli including both populations. The minimum accuracy in perceiving the inclusions with lower stiffness was found to be 52%, with the exception of the softest Dragon skin 20 material in NILS condition, whose rate decreased down to 33%. The global performance of identified inclusions, across all materials, showed an average of 63% and 60%, respectively, for ILS and NILS conditions (Figure 7).
(1)$$G\left(x\right)={\left[1+e^{-\frac{x-a}{b}}\right]}^{-1}$$


The CDF fitting function, introduced in Section 2 and reported in Equation (1), represents the probability of a correct response at a stiffness *x*, where *a* denotes the perceptual threshold, and *b* > 0 a scale parameter that affects the curve steepness. With respect to our previous study [43], the equation has been updated in order to account for an identification task rather than for a two-alternatives forced choice task.

The final evaluation of the perception, found by fitting the response datasets into the CDF, demonstrated that the user could distinguish an inclusion with stiffness higher than *a* = 2.5 N·mm^−1^ and *a* = 2.9 N·mm^−1^ in ILS and NILS conditions, respectively (Figure 7).

## 4. Discussion and Conclusions

This paper assesses the usability of the developed tactile system in tele-palpation to localize various stiffer polymeric nodules in the surrounding soft matrix during active exploration. The promising results demonstrate the ease and successful augmentation for navigation in a soft terrain. The four types of inclusions are observed to be paired with respect to their stiffness values (Softer: DS20, DS30; Stiffer: PDMS, SC) that are also reflected in the user’s accuracy graph.

The present study was conducted within two different telepresence conditions. Initially, in a more controlled environment, the sensing platform was placed near the user (In Line of Sight—ILS), while afterwards, the platform was moved to a remote location (Not In Line of Sight—NILS). Particularly, the latter condition presented two challenges: remote platform control and stiffness discrimination with the haptic glove. The intrinsic absence of vision of the platform in NILS with respect to ILS is displayed in the perceptual thresholds (*a* = 2.9 N·mm^−1^ versus *a* = 2.5 N·mm^−1^). Additionally, the network latency introduced a delay of the order of ms, demanding the subject’s attention on the active control, to coordinate the proprioceptive information with the moving platform, occasionally compromising attention to the sensory feedback. It was also observed that the subjects mostly took linear exploratory paths in both conditions, which was proven to be the best maneuver trajectory in the apparatus characterization (as reported in Table 1). We showed that the presented tactile telepresence system enabled the correct discrimination of the inclusions throughout the polymeric matrix, especially for the stiffer ones, in both ILS and NILS telepresence conditions. This work enriches the findings of our previous works [43,48,49,50], confirming that the adoption of spike-like stimulation, emulating the firing activity of skin mechanoreceptors, offers a usable language of feedback that can be delivered directly on the skin surface, to provide perceptual augmentation in tele-palpation with gestural active exploration.

Future works will be headed towards exploring the recognition of buried nodules along two directions: upgrading the number and realism of phantoms for medical applications, and enriching the mechanical information encoded. We will evaluate higher variations of material stiffness with the aim of discriminating healthy tissues from tumors, which are generally stiffer than the surroundings [57,58,59]. Furthermore, we will also include the evaluation of encoded feedback for both normal and shear components of the contact force through the neuromorphic model. This enrichment is expected to lead to a more detailed appreciation of stiffness in cases of anisotropic mechanical behavior of biomaterials.

## Figures and Tables

**Figure 1 sensors-19-00641-f001:**
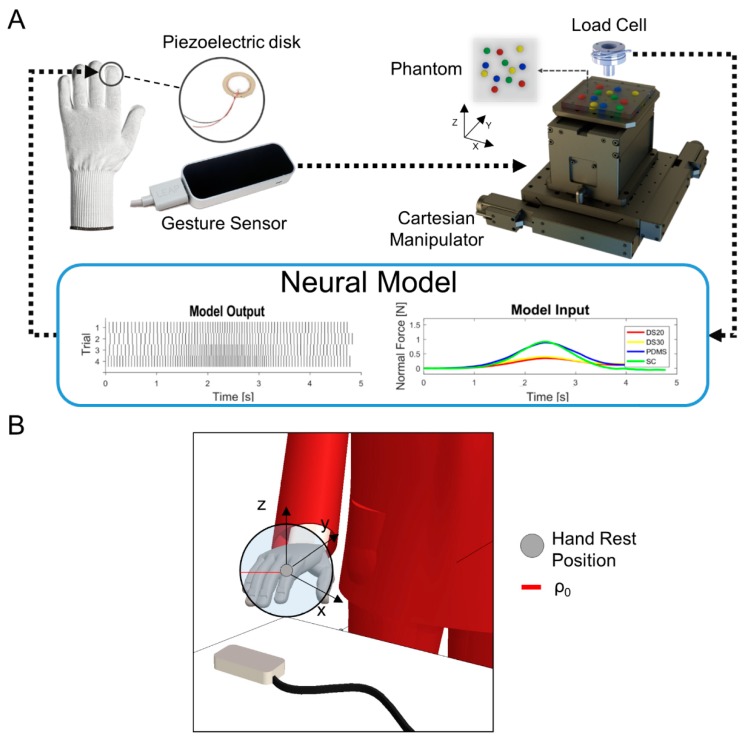
Experimental apparatus. (**A**) Left: the haptic sub-system comprising a textile glove with a detail of the encapsulated piezoelectric disk for index fingertip vibro-tactile stimulation and optical sensor for hand gesture tracking; right: tactile sub-system comprising a 3-axis cartesian manipulator with load cell and the indenter, and a detail on the silicon phantom displaying nodules placement. The two sub-setups were spatially separated to achieve In Line of Sight (ILS) and Not In Line of Sight (NILS) telepresence conditions. The plots at the bottom of the figure show the neural encoding of the normal force arising during the sliding phase of the phantom into spike trains for all the polymers, from the softest (red) to the hardest (green). (**B**) Details about the reference coordinates of the gesture sensor, highlighting the rest position and volume.

**Figure 2 sensors-19-00641-f002:**
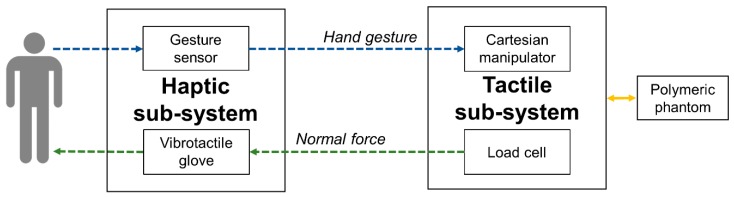
Block diagram: bidirectional data streaming between the haptic and tactile sub-systems provided via UDP: the optical controller conveyed speed and position of the center of mass of the user’s hand from the first environment to the cartesian manipulator in the remote one (blue arrow) and, while sliding, normal force data collected by the load cell from the platform to the vibro-tactile glove to deliver the spike-based stimulation (green arrow).

**Figure 3 sensors-19-00641-f003:**
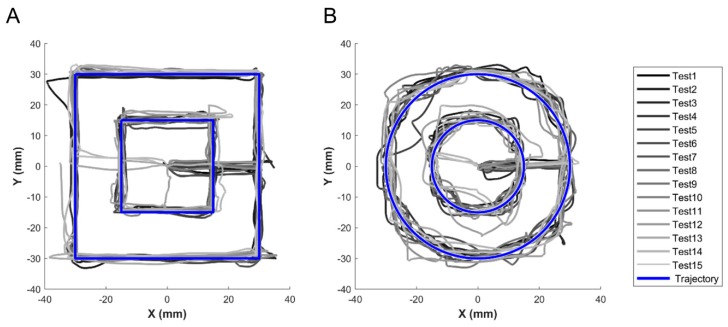
Graph highlighting target trajectories (blue lines) and tracked trajectories (gray lines) within the X–Y cartesian plane: (**A**) square shaped; (**B**) circular shaped. The gray lines starting at the origin represent the path to reach the target trajectories.

**Figure 4 sensors-19-00641-f004:**
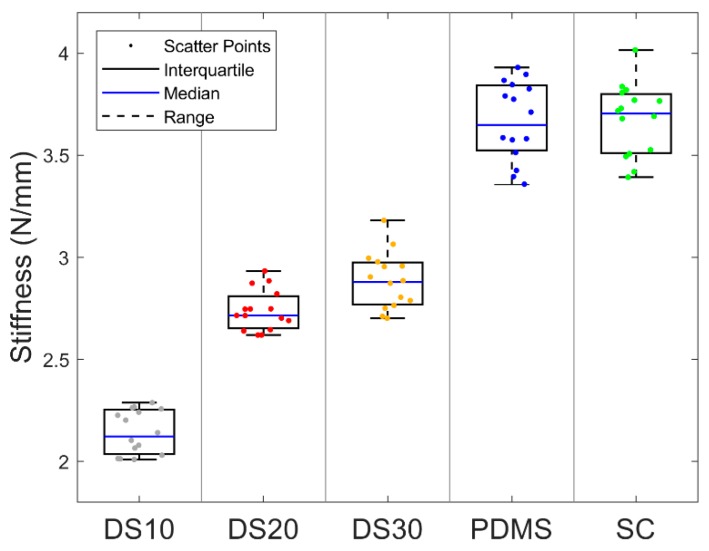
Mechanical characterization of the phantom: scatter points are the stiffness values collected for each material (Dragon Skin 10—DS10; Dragon Skin 20—DS20; Dragon Skin 30—DS30; polydimethylsiloxane—PDMS; Sorta Clear 40—SC) across the five trials of indentation; boxes represent interquartile ranges for the five materials; blue lines show the median values and black dashed lines the full ranges among the measured values.

**Figure 5 sensors-19-00641-f005:**
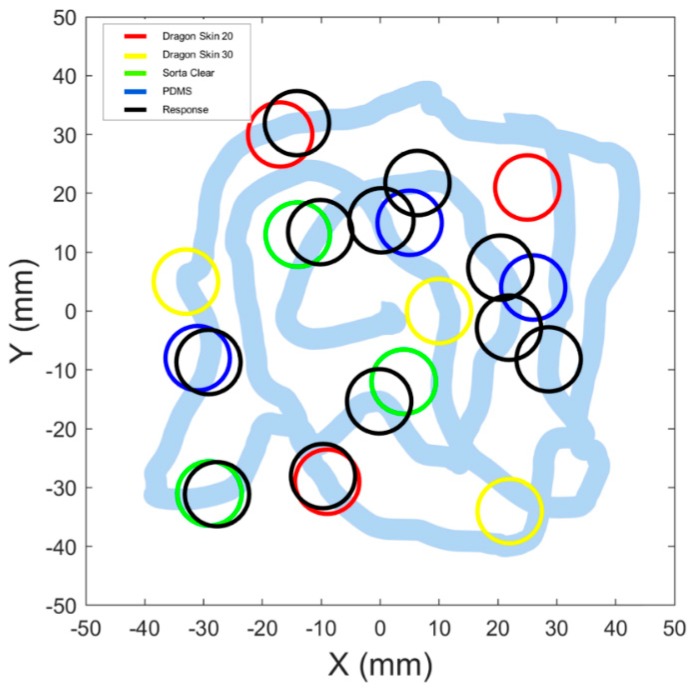
Example of responses given by a participant across an experimental session: colored circles mark the position of the inclusion set; black circles represent the position of the indenter when the subject pressed the key, and light blue line represents the trail of the probe on the phantom surface.

**Figure 6 sensors-19-00641-f006:**
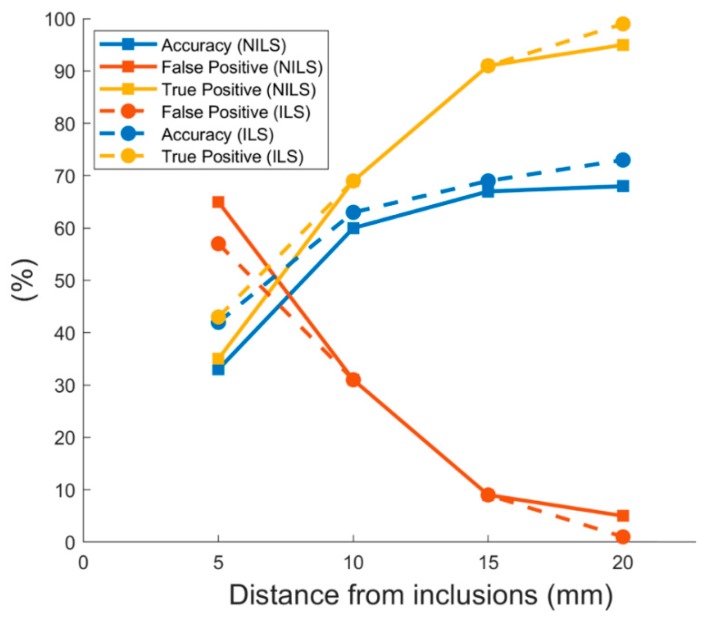
Tolerance of recognized inclusions: each point (dots for ILS and squares for NILS) represents the mean accuracy (blue line), mean false positive (red line), and mean true positive (yellow line) responses evaluated through a classification based on the admitted center-to-center distance between perceived inclusions and the real ones.

**Figure 7 sensors-19-00641-f007:**
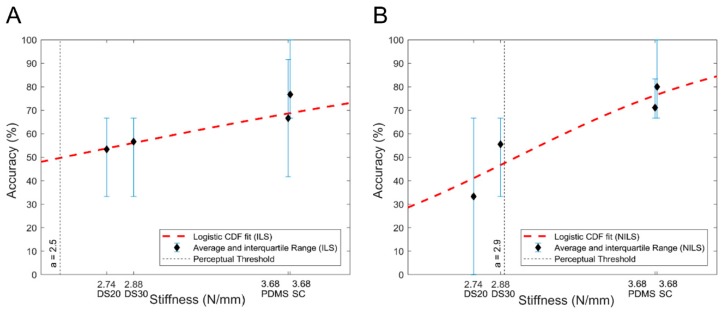
Psychometric curves for the psychophysical experiments: (**A**) ILS telepresence condition; (**B**) NILS telepresence condition. Black diamonds show the identification rate for all the encapsulated materials (average across participants); error bars are the interquartile range across participants; red dashed lines represent the logistic cumulative distribution function (CDF) fit.

**Table 1 sensors-19-00641-t001:** Results from characterization of the apparatus.

n = 15	Square (s_1_ = 60 mm)	Square (s_2_ = 30 mm)	Circle (r_1_ = 30 mm)	Circle (r_2_ = 15 mm)
**Target Area (mm^2^)**	3600	900	2827.43	706.86
**|Tracked Area – Target Area| (mm^2^) (µ ± σ)**	74.05 ± 65.79	43.27 ± 40.99	143.90 ± 121.28	91.74 ± 89.82
**Error Rate (%) (µ ± σ)**	2.01 ± 1.83	4.81 ± 4.55	5.09 ± 4.29	12.98 ± 12.71

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
