# Peer review of "Haptic Glove and Platform with Gestural Control For Neuromorphic Tactile Sensory Feedback In Medical Telepresence †"

_sensors, 2019, doi:10.3390/s19030641_

Round 1
Reviewer 1 Report
This paper presents a functional tele-palpation platform based on haptic feedback. Such platform consists of two modules: haptic and tactile. The haptic one comprises a glove with a vibrating piezo and an optical sensor. The tactile module involves a XYZ Cartesian manipulator, a load cell, an indenter, and a silicon phantom. The system allows users to control the displacement of the Cartesian manipulator and the indenter through hand gestures detected by the optical sensor. When the user touches an object in the tactile module, the load cell encodes the normal force into spike trains that make vibrate the piezo in glove. This way, the user is capable of perceiving object stiffness. Two telepresence modes were tested: In Line of Sight (users close to the platform) and Not In Line Sight (users 50 km apart from the platform). Results show that the system successfully delivers an appropriate encoding of the tele-palpated objects, it shows that users are capable of discriminating objects with different stiffness and that the NILS condition exhibits a different perceptual threshold due to the lack of vision and network latency.
Overall, this paper addresses an interesting topic on haptics, tele-operation platforms and stiffness sensing. The paper is well written, it is easy to read, and understand. I recommend its acceptance with the following minor remarks:
A) Content:
1) Lines 143-149: It is not clear the location of the optical sensor (distance to the user) and its sensing range. An image of the virtual sphere with all its dimensions would be helpful.
2) Lines 155-157: Parameters F_th, A, B, C,… need to be detailed. What do they stand for? They are further used in the very brief explanation of the Izhikevich spiking model (eqs (1) to (4)).
B) Style:
1) Figure 1: I recommend inverting the bottom plots (Model input and Model output) so that they are in agreement with your arrows. Model input between the Cartesian manipulator and the neural model, Model output between the neural model and the glove.
2) Line 142: results of such a characterization ARE reported in Section 3.
Author Response
Reviewer 1
Authors. We thank the reviewer for the careful and positive evaluation of the paper, and for the requested amendments that help enhancing the scientific quality and clarity of our study. The manuscript has been corrected taking into account the indications provided by the reviewer in the decision letter. Please find below a point by point reply to the comments made by the reviewer.
Reviewer. Comments and Suggestions for Authors
This paper presents a functional tele-palpation platform based on haptic feedback. Such platform consists of two modules: haptic and tactile. The haptic one comprises a glove with a vibrating piezo and an optical sensor. The tactile module involves a XYZ Cartesian manipulator, a load cell, an indenter, and a silicon phantom. The system allows users to control the displacement of the Cartesian manipulator and the indenter through hand gestures detected by the optical sensor. When the user touches an object in the tactile module, the load cell encodes the normal force into spike trains that make vibrate the piezo in glove. This way, the user is capable of perceiving object stiffness. Two telepresence modes were tested: In Line of Sight (users close to the platform) and Not In Line Sight (users 50 km apart from the platform). Results show that the system successfully delivers an appropriate encoding of the tele-palpated objects, it shows that users are capable of discriminating objects with different stiffness and that the NILS condition exhibits a different perceptual threshold due to the lack of vision and network latency.
Overall, this paper addresses an interesting topic on haptics, tele-operation platforms and stiffness sensing. The paper is well written, it is easy to read, and understand.
Authors. We gratefully thank the reviewer for the positive consideration of our work. The following replies provide evidences of the revisions made in order to integrate the constructive amendments suggested by the reviewer to enhance the description of methods, contents and style.
Reviewer. Lines 143-149: It is not clear the location of the optical sensor (distance to the user) and its sensing range. An image of the virtual sphere with all its dimensions would be helpful.
Authors. We thank the reviewer for the useful comment, which prompted us to improve the clarity of the description by revising Figure 1 accordingly and adding the following sentence:
“The used infrared light-based gesture sensor has a field of view of about 150 degrees and a range between 25 and 600 mm above the device, and it shows proper performance when it has a high-contrast view of the hand to be detected and tracked. Therefore, the device was placed just below and close to the rest position of the user’s hand and the workspace was set according to the specifications of the gesture sensor. Details about the reference coordinates of the gesture sensor, highlighting the rest position and volume are shown in Figure 1B.”
Reviewer. Lines 155-157: Parameters F_th, A, B, C… need to be detailed. What do they stand for? They are further used in the very brief explanation of the Izhikevich spiking model (eqs (1) to (4)).
Authors. We thank the reviewer for the helpful remark about constants and parameters of the spiking model mentioned above. Since the explanation of the model goes beyond the objectives of this work, and according with the suggestion of another reviewer to remove equations 1-4 because they are not needed to understand the work, we eliminated them from Section 1. We discussed the Izhikevich model in the revised manuscript and reported references [34-35] for more detailed information about the equations, parameters (a-d) and constants (A-C) of the employed spiking model.
Reviewer. Figure 1: I recommend inverting the bottom plots (Model input and Model output) so that they are in agreement with your arrows. Model input between the Cartesian manipulator and the neural model, Model output between the neural model and the glove.
Authors. We thank the reviewer for the recommendation and we revised Figure 1 accordingly. Please note that the revisions made to Figure 1 now should allow a better understanding of information flow between the sub-systems of the developed system.
Reviewer. Line 142: results of such a characterization ARE reported in Section 3.
Authors. We thank the reviewer for the careful revision. The text has been amended accordingly.

Reviewer 2 Report
The paper is written well and it is easy to follow the content.
As the current article is an extended version of the paper published in the 2018 IEEE International Symposium Medical Measurements, I would recommend the authors to improve their introduction and related work parts. You can split the introduction part in three I) motivation and problem definition II) related work and state-of-the-art III) Your contribution in this article.
I strongly recommend to describe some of the state-of-the-art research more in detailed instead of writing a general paragraph and cite a bunch of papers at the end of the paragraph.
For instance, you cited 10 papers at the end of the line 56 (paragraph 49-56) without giving any description. Or in line 70, you wrote about active exploration without any citation. I would recommend to give a bigger picture about the application of the tactile sensing in different fields. It helps your article to be more interesting for more researches. For example:
Tactile- perception in Human
Tactile sensibility in the human hand: relative and absolute densities of four types of mechanoreceptive units in glabrous skin.RS Johansson, AB Vallbo The Journal of physiology 286 (1), 283-300
Roles of glabrous skin receptors and sensorimotor memory in automatic control of precision grip when lifting rougher or more slippery objectsRS Johansson, G Westling Experimental brain research 56 (3), 550-56
Tactile Sensing Technology
“New materials and advances in making electronic skin for interactive robots Advanced Robotics, 29 (11), pp. 1359-1373, 2015
“A robust, low-cost and low-noise artificial skin for human-friendly robots,” in Proc. IEEE Int. Conf. Robot. Autom., 2010, pp. 4836–4841.
Tactile-based material recognition
"Learning dynamic tactile sensing with robust vision-based training." IEEE transactions on robotics 27, no. 3 (2011) 545-557.
Robust tactile descriptors for discriminating objects from textural properties via artificial robotic skin, IEEE Transactions on Robotics, 34(4), pp. 985-1003, 2018
Tactile based object manipulation and slip detection
estimation and slip detection/ classification for grip control using a biomimetic tactile sensor,.in IEEE International Conference on Humanoid Robots, pp. 297.303,2015.
Detection and prevention of slip using sensors with different properties embedded in elastic artificial skin on the basis of previous experience, Robotics and Autonomous Systems,vol. 62, no. 1, pp. 46.52, 2014.
Tactile-based Object Center of Mass Exploration and Discrimination"EEE International Conference on Humanoids Robot (Humanoids), 2017
Passive vs Active tactile exploration
Tactile-based active object discrimination and target object search in an unknown workspace Autonomous Robots 42 (3), 1573-7527, 2018
Active Prior Tactile Knowledge Transfer for Learning Tactual Properties of New Objects
Sensors 18 (2), 634, 2018
Active tactile transfer learning for object discrimination in an unstructured environment using multimodal robotic skin International Journal of Humanoid Robotics (IJHR) 15 (1), pp. 1-20, 2017
Author Response
Reviewer 2
Authors. We thank the reviewer for the careful and positive evaluation of the paper, and for the recommendations that help enhancing the scientific quality and clarity of our study. The manuscript has been corrected taking into account the indications provided by the reviewer in the decision letter. Please find below a point by point reply to the comments made by the reviewer.
Reviewer. Comments and Suggestions for Authors
The paper is written well and it is easy to follow the content.
As the current article is an extended version of the paper published in the 2018 IEEE International Symposium Medical Measurements, I would recommend the authors to improve their introduction and related work parts. You can split the introduction part in three I) motivation and problem definition II) related work and state-of-the-art III) Your contribution in this article.
I strongly recommend to describe some of the state-of-the-art research more in detailed instead of writing a general paragraph and cite a bunch of papers at the end of the paragraph. For instance, you cited 10 papers at the end of the line 56 (paragraph 49-56) without giving any description. Or in line 70, you wrote about active exploration without any citation. I would recommend to give a bigger picture about the application of the tactile sensing in different fields. It helps your article to be more interesting for more researches.
For example:
Tactile- perception in Human
Tactile sensibility in the human hand: relative and absolute densities of four types of mechanoreceptive units in glabrous skin. RS Johansson, AB Vallbo. The Journal of physiology 286 (1), 283-300
Roles of glabrous skin receptors and sensorimotor memory in automatic control of precision grip when lifting rougher or more slippery objects. RS Johansson, G Westling Experimental brain research 56 (3), 550-56
Tactile Sensing Technology
“New materials and advances in making electronic skin for interactive robots Advanced Robotics, 29 (11), pp. 1359-1373, 2015
“A robust, low-cost and low-noise artificial skin for human-friendly robots,” in Proc. IEEE Int. Conf. Robot. Autom., 2010, pp. 4836–4841.
Tactile-based material recognition
"Learning dynamic tactile sensing with robust vision-based training." IEEE transactions on robotics 27, no. 3 (2011) 545-557.
Robust tactile descriptors for discriminating objects from textural properties via artificial robotic skin, IEEE Transactions on Robotics, 34(4), pp. 985-1003, 2018
Tactile based object manipulation and slip detection
estimation and slip detection/ classification for grip control using a biomimetic tactile sensor,.in IEEE International Conference on Humanoid Robots, pp. 297.303,2015.
Detection and prevention of slip using sensors with different properties embedded in elastic artificial skin on the basis of previous experience, Robotics and Autonomous Systems,vol. 62, no. 1, pp. 46.52, 2014.
Tactile-based Object Center of Mass Exploration and Discrimination" IEEE International Conference on Humanoids Robot (Humanoids), 2017
Passive vs Active tactile exploration
Tactile-based active object discrimination and target object search in an unknown workspace
Autonomous Robots 42 (3), 1573-7527, 2018
Active Prior Tactile Knowledge Transfer for Learning Tactual Properties of New Objects. Sensors 18 (2), 634, 2018
Authors. We gratefully thank the reviewer for the very helpful remarks. The introduction has been partitioned and extended accordingly.

Reviewer 3 Report
The paper presents some results from a set of experiments that consist in exploring objects of different stiffness embedded in a phantom. In the abstract, the authors say they present “a novel telepalpation apparatus” and they assess it with psychophysical tests.
As a general comment, the authors refer to their previous work in the reference [34] of their paper, where they report studies about the recognition of objects with different stiffness. The material they add this time is the development of a closed loop control where a camera is used to track the movements of the hand of the user and the motorized platform is moved accordingly. Then, this set up is used in an experiment where the user virtually explores the phantom to find the buried objects. Since the neuromorphic model and their performance in transmitting the information of the mechanical properties were already reported in the previous work, this material should be removed (for instance equations (1)-(4), that are not needed to understand the work and can be found in [34]). Moreover, since the key point is the closed control loop, it should be described and characterized in more detail. I mean that the results of the paper are given for a given tolerance between the actual and the perceived distance. However, there is no data about the uncertainty of the system, i.e. the error of the platform in tracking the hand (static and dynamic). In the way the results are given in the paper, it is not possible to evaluate if the limitations in performance are due to the limitations of the system or to psychophysical reasons. In summary, since you say that you present the apparatus, you must describe it in detail and provide data about its performance to give all the information to the reader.
A second general comment is related to the results in section 3. Since it is the main contribution of the paper, I think the authors should provide results from a much wider range of objects. They are only four and two of them have very similar stiffness. Therefore, the conclusions in Figure 6 (for instance the perception thresholds) are not so reliable. Moreover, I wonder why you use a different fitting curve (equation 5 of your paper) that in [34]. You should provide a reference for the equation (5).
In summary, I think this work is not mature yet. Since there is work behind and could provide interesting information, I would recommend “major revision”, but I think a few months will be needed before it is ready to be published, in my opinion, so I recommend “reject” but I encourage the authors to complete it and send it again as a new submission.
Minor comments:
Style:
- Remove material already reported in [34] and not needed to understand the paper.
- Refer to section 2 in the discussions of the results of 3, to explain how they are obtained (for instance the use of equation (5))
Minor:
- Size of the labels in the figures too small sometimes
- Explain the meaning of DS10, DS20,…SC in the caption of Figure 3
Author Response
Reviewer 3
Authors. We thank the reviewer for the careful evaluation of the paper, and for the requested amendments that help enhancing the scientific quality and clarity of our study. The manuscript has been corrected taking into account the indications provided by the reviewer in the decision letter. Please find below a point by point reply to the comments made by the reviewer.
Reviewer. Comments and Suggestions for Authors
The paper presents some results from a set of experiments that consist in exploring objects of different stiffness embedded in a phantom. In the abstract, the authors say they present “a novel telepalpation apparatus” and they assess it with psychophysical tests.
As a general comment, the authors refer to their previous work in the reference [34] of their paper, where they report studies about the recognition of objects with different stiffness. The material they add this time is the development of a closed loop control where a camera is used to track the movements of the hand of the user and the motorized platform is moved accordingly. Then, this set up is used in an experiment where the user virtually explores the phantom to find the buried objects. Since the neuromorphic model and their performance in transmitting the information of the mechanical properties were already reported in the previous work, this material should be removed (for instance equations (1)-(4), that are not needed to understand the work and can be found in [34]).
Authors. We thank the reviewer for the comment. Accordingly, we removed the equations of the neural model from the text and we added the references for further details and explanations. The text of the pertinent paragraph now is as follows:
“The measured force was sent to the haptic sub-system to be encoded into spike patterns, that triggered the piezoelectric actuator. We implemented a neuromorphic feedback strategy based on a regular Izhikevich spiking model discretized using Euler’s method at 5 kHz [53]. The chosen model efficiently encodes the temporal dynamics of a mechanoreceptor including the biological plausibility of computationally intense Hodgkin-Huxley model [54] and the computational efficiency of integrate-and-fire model [55].
The neuromorphic activation of the transducer was achieved by setting the input to the neuron proportional to the magnitude of the normal force measured by the remote subsystem. Further details about the neural model and definition of its parameters can be found in our previous works [48,56]. Initial calibrations were also performed to counterbalance the effect of saturation of the neural model [48].
The spikes generated by the Izhikevich model were sent to the vibro-tactile glove by means of a piezoelectric driver (DRV2667, Texas Instruments). This driver facilitated in setting the actuation parameters in analog mode to have a gain of 40.7 dB, 200 V peak-to-peak voltage amplitude, and 105 V offset voltage.”.
Reviewer. Moreover, since the key point is the closed control loop, it should be described and characterized in more detail. I mean that the results of the paper are given for a given tolerance between the actual and the perceived distance. However, there is no data about the uncertainty of the system, i.e. the error of the platform in tracking the hand (static and dynamic). In the way the results are given in the paper, it is not possible to evaluate if the limitations in performance are due to the limitations of the system or to psychophysical reasons. In summary, since you say that you present the apparatus, you must describe it in detail and provide data about its performance to give all the information to the reader.
Authors. We thank the reviewer for the useful comment, that prompted us performing additional experiments to provide a quantitative characterization of the platform. Five subjects (4 men and 1 woman between 28 and 30 years of age), enrolled among the staff of The BioRobotics Institute of Sant’Anna School of Advanced Studies, took part in the experiment. They were asked to drive the stages of the platform in order to perform a set of target trajectories within the X-Y cartesian plane. The target trajectory and the subject’s relative position on this plane were part of the GUI providing the visual feedback in the haptic sub-system. The target trajectories consisted of 2 square-shaped (s1= 60 mm and s2 = 30 mm, in length) and 2 circular (r1 = 30 mm and r2 = 15 mm, in radius) geometries, that had to be followed 3 times by each subject, starting from their centers. Deviations in the trajectories, used to evaluate the system in terms of performance, were calculated using the area comprised between the perimeter of the target and executed trajectories, by means of the boundary function (MATLAB, Mathworks). We re-organized the sectioning of materials and methods and results in order to better explain the experimental setup, the platform and inclusions characterization protocols and results, and the inclusions identification experimental methods and results. Furthermore, a new Figure (named Figure 3 in the revised manuscript) and a Table have been added with quantitative data.
All these revisions affect the organization and text of Sections 2.2 and 2.3 in the Materials and Methods and Sections 3.1 and 3.2 of the revised manuscript.
Reviewer. A second general comment is related to the results in section 3. Since it is the main contribution of the paper, I think the authors should provide results from a much wider range of objects. They are only four and two of them have very similar stiffness. Therefore, the conclusions in Figure 6 (for instance the perception thresholds) are not so reliable. Moreover, I wonder why you use a different fitting curve (equation 5 of your paper) that in [34]. You should provide a reference for the equation (5). In summary, I think this work is not mature yet. Since there is work behind and could provide interesting information, I would recommend “major revision”, but I think a few months will be needed before it is ready to be published, in my opinion, so I recommend “reject” but I encourage the authors to complete it and send it again as a new submission.
Authors. We thank the reviewer for the useful comment. The limitation with the range of evaluated inclusions has been stated in the Discussion and Conclusions as a perspective for future works. Furthermore, the difference with respect to our previous study has been discussed to justify Equation 5 (current Equation 1), as follows:
“The CDF fitting function, introduced in Section 2 and reported in Eq. (1), represents the probability of a correct response at a stiffness x, where a denotes the perceptual threshold, and b > 0 a scale parameter that affects the curve steepness. With respect to our previous study [47], the equation has been updated in order to account for an identification task rather than for a two-alternatives forced choice task.”.
We hope that the reviewer can appreciate the efforts done in revising the paper according to his/her constructive comments.
Reviewer. Style: Remove material already reported in [34] and not needed to understand the paper.
Authors. We thank the reviewer for the helpful comment and we revised the manuscript accordingly.
Reviewer. Style: Refer to section 2 in the discussions of the results of 3, to explain how they are obtained (for instance the use of equation (5)).
Authors. We thank the reviewer for the useful suggestion. Accordingly, we added the reference to Section 2 to clarify how the “a” perceptual thresholds have been obtained. We also rephrased previous L.202-204 as follows:
“A logistic fitting, using a Cumulative Distribution Function (CDF) [57] and the nlinfit function of Matlab MATLAB, was performed for each material to carry out the rate of correct perception and the perceptual thresholds for both the investigated experimental conditions.”
Reviewer. Minor:
- Size of the labels in the figures too small sometimes
- Explain the meaning of DS10, DS20,…SC in the caption of Figure 3.
Authors. We thank the reviewer for the useful remarks, which prompted us to improve both font size of labels across the figures and the clarity of the caption of Figure 3. Accordingly, this caption has been changed as follows:
“Figure 4. Mechanical characterization of the phantom: scatter points are the stiffness values collected per each material (Dragon Skin 10 - DS10; Dragon Skin 20 - DS20; Dragon Skin 30 - DS30; PDMS; Sorta Clear 40 - SC) across the five trials of indentation; boxes represent interquartile ranges for the five materials; blue lines show the median values and black dashed lines the full ranges among the measured values.”

Round 2
Reviewer 3 Report
My comments to the authors have been addressed. Thank you